# Potential Role of Phytochromes A and B and Cryptochrome 1 in the Adaptation of *Solanum lycopersicum* to UV-B Radiation

**DOI:** 10.3390/ijms241713142

**Published:** 2023-08-24

**Authors:** Anna Abramova, Mikhail Vereshchagin, Leonid Kulkov, Vladimir D. Kreslavski, Vladimir V. Kuznetsov, Pavel Pashkovskiy

**Affiliations:** 1K.A. Timiryazev Institute of Plant Physiology, Russian Academy of Sciences, Botanicheskaya Street 35, Moscow 127276, Russia; ann.kiedis2000@gmail.com (A.A.); mhlvrh@mail.ru (M.V.); vlkuzn@mail.ru (V.V.K.); pashkovskiy.pavel@gmail.com (P.P.); 2Department of Technologies for the Production of Vegetable, Medicinal and Essential Oils, Russian State Agrarian University, Moscow Timiryazev Agricultural Academy, Timiryazevskaya Street 49, Moscow 127550, Russia; q-q639@mail.ru; 3Institute of Basic Biological Problems, Russian Academy of Sciences, Institutskaya Street 2, Pushchino 142290, Russia

**Keywords:** tomato, photoreceptor mutants, stress resistance, photosynthesis, UV-B

## Abstract

UV-B causes both damage to the photosynthetic apparatus (PA) and the activation of specific mechanisms that protect the PA from excess energy and trigger a cascade of regulatory interactions with different photoreceptors, including phytochromes (PHYs) and cryptochromes (CRYs). However, the role of photoreceptors in plants’ responses to UV-B radiation remains undiscovered. This study explores some of these responses using tomato photoreceptor mutants (*phya*, *phyb1*, *phyab2*, *cry1*). The effects of UV-B exposure (12.3 µmol (photons) m^−2^ s^−1^) on photosynthetic rates and PSII photochemical activity, the contents of photosynthetic and UV-absorbing pigments and anthocyanins, and the nonenzymatic antioxidant capacity (TEAC) were studied. The expression of key light-signaling genes, including UV-B signaling and genes associated with the biosynthesis of chlorophylls, carotenoids, anthocyanins, and flavonoids, was also determined. Under UV-B, *phyab2* and *cry1* mutants demonstrated a reduction in the PSII effective quantum yield and photosynthetic rate, as well as a reduced value of TEAC. At the same time, UV-B irradiation led to a noticeable decrease in the expression of the *ultraviolet-B receptor (UVR8), repressor of UV-B photomorphogenesis 2 (RUP2), cullin 4 (CUL4), anthocyanidin synthase (ANT), phenylalanine ammonia-lease (PAL)*, and *phytochrome B2* (*PHYB2)* genes in *phyab2* and *RUP2, CUL4, ANT, PAL*, and *elongated hypocotyl 5 (HY5)* genes in the *cry1* mutant. The results indicate the mutual regulation of UVR8, PHYB2, and CRY1 photoreceptors, but not PHYB1 and PHYA, in the process of forming a response to UV-B irradiation in tomato.

## 1. Introduction

Ultraviolet (UV) radiation is an important environmental factor that significantly affects the photosynthetic parameters and productivity of plants. Plants grow under conditions of significant fluctuations in the intensity and spectral characteristics of sunlight. They perceive the quality of light through several main photoreceptors: phytochromes sensitive to red and far red light (620–750 nm); cryptochromes, phototropins, and Zeitlupe proteins sensitive to blue light and UV-A radiation (315–500 nm); and the UV-B receptor UVR8 [1]. Ultraviolet B (UV-B) (280–315 nm) and ultraviolet A (UV-A) (315–400 nm) radiation comprise small portions of solar radiation but regulate many aspects of plant development, physiology, and metabolism [2]. The effects of their exposure strongly depend on the type of plant, the dose of radiation, and the degree of plant acclimatization. UV-B radiation can act as a stressor or photomorphogenetic factor depending on its intensity and duration of exposure [3,4,5]. Low doses of UV-B trigger photomorphogenetic responses, such as suppressing stem elongation and stomatal opening and stimulating the accumulation of UV-B-absorbing flavonoids [6,7,8]. Moreover, UV-B can reduce the adverse effects of other stresses, such as drought [9], by increasing cellular water content, increasing photosynthesis, and enhancing tissue antioxidant capacity [10,11,12]. It has been shown that exposure to UV-B can lead to an increase in the photosynthesis rate, stomatal conductance, and water-use efficiency (WUE), which ultimately has a positive effect on plant productivity [13,14]. In addition, higher doses of UV-B can reduce the rate of CO_2_ assimilation by closing stomata and reducing Rubisco activity [5]. During seed germination, UV-B can also increase the content of L-ascorbic acid and polyphenols, and the antioxidant activity and the content of beta-carotene and lycopene [15]. Oxidative stress caused by UV-B initiates protective reactions leading to an increase in phenolic compounds and antioxidants [16,17,18]. As a rule, in this case, stimulation of phenylalanine ammonia-lyase (PAL) [19], chalcone synthase (CHS), and flavonol synthase (FLS) is observed. This leads to increased levels of polyphenols [20] and stimulation of the formation of some volatile organic compounds [21] and alkaloids, which are generally beneficial for human nutrition [16].

UV-B (280–315 nm) can reduce plant growth and inhibit photosynthesis by generating reactive oxygen species that can cause oxidative damage to membrane lipids, nucleic acids, and proteins [22]. The thylakoid membrane is extremely susceptible to UV-B radiation, which leads to a decrease in the functional activity of chloroplasts [23]. UV radiation is absorbed by carotenoids, porphyrins, and quinones, leading to destabilization and functional impairment of the major macromolecules associated with them [24]. The degradation of chlorophylls and carotenoids has been described as a typical symptom of UV-B stress [25]. Although UV-B radiation influences many physiological processes in plants, PSII is usually indicated as the main target of UV-B action [26]. PSII reaction centers contain the main proteins D1 and D2, which can be damaged by light; however, the repair and synthesis of these proteins under normal light conditions are quite fast. Upon exposure to UV-B, the degradation rate of D1 and D2 increases significantly [27], leading to an imbalance of these proteins and, as a result, impaired PSII functioning [26].

There is cross-signaling between phytochromes, cryptochromes, and UVR8, which indicates the existence of a regulatory network that, in addition to the photoreceptors themselves, includes components of light and hormonal signaling, transcription factors, second messengers, and ROS. Previously, reduced *UVR8* expression was observed in response to UV-B in the *Arabidopsis thaliana hy4 (cry1)* mutant [28]. This suggests a link between CRY1, UVR8, and UV-B. The photoreceptors UVR8 and phytochrome B (PHYB) cooperate to improve plant growth and adaptation potential [29]. Among phytochromes, PHYA and PHYB play a key role, the physicochemical properties of which are most fully characterized [30,31]. PHYA and PHYB are involved in many PA responses to oxidative stress caused by various environmental factors, including UV-B. Increasing the content of PHYA and PHYB enhances the resistance of PA to oxidative stress, and the deficiency of these PHYs can reduce resistance [32]. Among photoreceptors, CRY1 also plays an important role in PA stress resistance [33] since it, similar to PHYs, is able to activate the transcription of genes that restore or protect photosystems [34]. Cryptochromes can influence stomatal aperture, the regulation of which is necessary for the maintenance of water status and facilitation of CO_2_ intake under both normal and stress conditions [35]. The abovementioned results indicate that PHYs and CRYs play a central role in the regulation of the expression of a large number of photosensitive genes involved in the regulation of a wide range of processes, from photomorphogenesis to stress reactions [36]. PHY signaling under the action of red light induces the expression of genes that overlap significantly with genes induced by UV radiation and blue light [5]. These co-induced genes are associated with plant adaptation to UV-B (such as *HY5, MYBs, E3 ubiquitin-protein ligase COP1*, and *phytochrome interacting factors PIFs*). Additionally, there is interaction among photoreceptors upon their action on genes *CHS, ANT*, and *PAL*, regulating the activity of enzymes that are important for the synthesis of various phenolic compounds and metabolites acting as protective molecules [37,38]. The presented data also indicate that a significant portion of light reactions, which are directly regulated by PHYs, CRYs, and related genes, can also be carried out under the action of UV radiation.

Under natural environmental conditions, sunlight activates all photoreceptors because it contains an extremely wide spectrum of radiation. At the same time, the spectral composition of light can differ at sunrise and sunset, and the intensity of the ultraviolet radiation is related to the intensity of sunlight and the height above sea level where the plants grow. In this regard, we studied photoreceptor mutants to understand how they are involved in light signaling under conditions of additional long-term UV-B irradiation. In this regard, the main goal of our work was to study the effect of UV-B radiation on various tomato mutants with deficiencies in PHYA, PHYB, and CRY1. We tried to show how the deficiency of photoreceptors affects the primary photochemical processes of photosynthesis, net photosynthetic and transpiration rates, content and composition of the main photosynthetic and other pigments with antioxidant properties, and the ability to absorb UV-B. We hypothesized that tomato photoreceptor mutants are able to respond differently to UV-B irradiation due to the presence of cross-signaling among different photoreceptors. We also suggest that PHYB2 and CRY1 play a key role in UV-B adaptation in *S. lycopersicum*.

## 2. Results and Discussion

### 2.1. Content of Pigments

Mutants and WT plants differed in their chlorophyll and carotenoid contents. Thus, in the *phyab2* mutant, the content of these pigments was somewhat lower than that in the *cry1* mutant. However, after UV-B irradiation, the content of pigments in all mutants decreased by 1.5–2 times (Table 1).

Initially, the anthocyanin content was highest in the WT and lowest in the *cry1* mutant. After irradiation with UV-B, the content of anthocyanins increased in all mutants by 2–4 times, with the exception of *phyb1*, in which the content changed minimally (Table 1).

The UAP content was the highest in the *phyb1* mutant, while in *phyab2* it was 1.6 times lower but higher than in other mutants. However, after exposure to UV-B, this difference among mutants was negligible. In addition, the UAP content in *cry1* was 1.5 times lower than that in *phyab2* (Table 1).

Initially, the TEAC value was the highest in the *phya* mutant and the lowest in *phyb1* (1.3 times lower than in the *phya*). After UV-B irradiation, the highest TEAC was observed in *phya*, while TEAC in *phyab2* and *cry1* mutants was 1.4–1.5 times lower than that in the *phya*. The *phyab2* and *phyb1* mutants showed intermediate TEAC values (Table 1).

### 2.2. Photosynthetic Activity, and Net Photosynthetic and Transpiration Rates

Initially, during the experiment, the portion of UV-B available in the spectrum had no effect on the PSII activity and photosynthesis rate of the WT (Table 2), which corresponded to relevant values indicated earlier in pine seedlings [39]. Such a level of UV-B likely does not affect photosynthetic parameters as much. Hence, to demonstrate the effects of UV-B, we needed to use higher doses of UV-B. The application of 10 µmol (photons) m^−2^s^−1^ for 1 and 2 h did not lead to inhibition of photosynthetic activity, but irradiation with UV-B for 16 h led to a small but reliable effect on PA activity. However, irradiation for 3 days led to changes in the activity of PSII and the net photosynthetic rate in some mutants.

UV-B irradiation demonstrated little effect on the maximum quantum yield of PSII in all studied plants (Table 2). The effective quantum yield of PSII Y(II) in *phyab2* and *cry1* decreased by two-fold relative to the initial point. For other mutants, the decrease was not noticeable (Table 2). After UV-B irradiation, a decrease in the non-photochemical quenching index (NPQ) was observed in WT (Table 2). It is worth noting that before irradiation, the NPQ value in the *phya* mutant was the lowest (Table 2). In the studied mutants, after UV-B irradiation, a 3–6-fold decrease in NPQ was observed (Table 2). Initially, the values of Y(NPQ) and Y(NO) practically did not differ. However, after irradiation, the Y(NPQ) value increased in the *phyab2* mutant, and the Y(NO) value increased in *cry1.* In other mutants, the changes were less pronounced (Table 2).

The net photosynthetic rate at the beginning of the experiment was slightly different among the mutants, but the most noticeable changes in the values were observed in the WT and *phyb1* mutant (Table 2). After UV-B irradiation, the lowest photosynthetic rates were observed in the *phyab2* mutant and *cry1,* in which the values of the rates (relative to the initial point) were 3.4 and 4.3 times lower, respectively (Table 2).

A decrease in transpiration (17 times) after exposure to 48 h of UV-B was found in *phyb1* (Table 2), and the smallest decrease was found in *phya* (2.6 times) (Table 2). It should be noted that in *cry1*, transpiration remained at the level of the initial point, while in the *phyab2* mutant, transpiration slightly increased (Table 2).

### 2.3. Transcript Levels of the Studied Genes

*SPA* transcripts were initially higher in all mutants than in the WT, except for the *cry1* mutant, in which *SPA* expression was decreased by almost 1.5 times relative to that in the WT. After UV-B irradiation, *SPA* expression decreased in all phytochrome mutants, but in *cry1, SPA* expression was not changed from the initial point (Figure 1).

*HY5* expression was at a level comparable to that of the WT in all mutants with the exception of *cry1,* where expression was reduced by almost two-fold. After UV-B exposure, *HY5* decreased in *phya* and the *phyab2* mutant, and *phyb1,* in contrast, showed a 50% increase in the transcript level. In *cry1,* the reduction was the most pronounced (Figure 1).

The expression of the *BBX21* gene was initially lower in the *phyb1, phyab2*, and *cry1* mutants than in the WT, and after UV-B irradiation, the same mutants showed an increase in the expression of this gene (by 50–80% relative to the WT) (Figure 1).

The expression of *CUL4* was initially reduced in the WT, *phyb1*, and *cry1* mutants compared to the *phyab2* mutant. However, after UV-B irradiation, *phyb1* showed an increase in the expression by more than three times relative to the initial point; in *phyab2,* on the contrary, there was a decrease in the expression of *CUL4* by more than four-fold relative to the initial point, and the mutant *cry1* showed no change in expression after UV-B irradiation.

The transcript level of the *HY2* gene in the mutants was higher than that in the WT, with the exception of *cry1.* After UV-B irradiation, a decrease in the expression of *HY2* was observed in all mutants relative to WT, while the largest decrease relative to the initial point was observed in *phya* and *cry1* (Figure 1).

*PIF4* expression was 2-fold lower relative to *WT* in the *phya* mutant, while the *phyab2* mutant expression of *PIF4* was 3.5-fold higher than in WT. After UV-B irradiation, *PIF4* gene expression increased in WT and was not very different among the studied mutants (Figure 1).

The *DET1* transcript level was initially slightly higher in the *phyab2* mutant and *cry1.* After irradiation, the *phyab2* mutant and *cry1* showed the lowest expression of this gene (Figure 1).

*COP1* expression was initially the highest in the *phyab2* mutant, and UV-B irradiation led to an almost three-fold decrease in the expression of this gene. In single phytochrome mutants, on the contrary, UV-B irradiation led to an increase in the expression of this gene by more than two times relative to the initial point, while, in *cry1,* expression was not changed.

The transcription level of *CRY1* was the highest in the *phya* and *phyb1* mutants but minimal in *cry1*. After UV-B irradiation, the expression of *CRY1* decreased in all mutants (Figure 1).

The expression of *CRY2* was initially the highest in the double and *cry1* mutants, and after UV-B irradiation, the same mutants showed a decrease in the expression of this gene (Figure 1).

The level of *PHYB1* transcripts was reduced in all mutants, and after irradiation, it increased in the mutants and was the lowest in the *phyab2* mutant (Figure 1).

Initially, the maximum expression of *UVR8* was in the *phyab2* mutant, and after irradiation, the level of transcripts of this gene decreased and was lower than that in other mutants.

At the initial point, the expression of *RUP2* was the highest in the *phyab2* mutant, and after UV-B irradiation, it increased in the *phya* and *phyb1* mutants but decreased in the *cry1* and *phyab2* mutants (Figure 1).

At the initial point, the expression level of the *psbA* gene was not different among the mutants and WT, and after UV-B irradiation, the transcript level of this gene decreased in all mutants.

The *psbD* transcription level was initially the highest in the *phyab2* mutant, and, when exposed to UV-B, it decreased in all mutants, but the maximum decrease was observed in *phyb1* and *phyab2* mutants (Figure 1).

The *CAB1* gene before irradiation was maximized in *phyb1* and *phyab2*, and after UV-B irradiation, in *phya,* it increased more than two-fold, and in *cry1,* it was increased.

The expression level of the *PSBS* gene at the initial point was the highest in the *phya, phyb1,* and *phyab2* mutants, and after irradiation, it decreased in all the studied mutants, although the maximum decrease was observed in *cry1* (Figure 1).

Initially, the expression of the *ANT* gene was high only in the double and *cry1* mutants, and after UV-B irradiation, the expression of this gene increased relative to the WT only in *phya* and *phyb1* by more than nine- and seven-fold, respectively. At the same time, *ANT* expression in *cry1* and *phyab2* decreased and was the lowest among the mutants.

*PAL* expression was the highest in the *phyab2* mutant, and after irradiation, the expression increased in all mutants except the *phyab2* mutant. In *phya* and *phyb1,* there was an increase compared to the WT (Figure 1).

*CHS* gene expression in all mutants was lower than that in WT, and after irradiation, the expression increased in all mutants, but an increase was found in *phyb1* (Figure 1).

### 2.4. Impact of UV-B and Photoreceptor Deficiency on Photosynthetic Processes

When researching the effects of UV-B on the activity of primary photochemical photosynthetic processes, PSII is considered a main target since it is one of the most sensitive to the UV-B component of PA [5]. In our experiments, PSII activity was evaluated as the value of Y(II) decreased under the action of UV-B (Table 2). A particularly noticeable decrease in the Y(II) value was observed in the *cry1* and *phyab2* mutants (Table 2). However, these and other mutants had high values of the maximum quantum yield, which indicates that UV-B has little effect on the primary processes of charge separation in the PSII reaction centers, but rather affects electron transfer to the plastoquinone pool and further along the electron transport chain [40]. At the same time, enhanced values of Y(NO) were observed in the *cry1* mutant and Y(NPQ) in the *phyab1* mutant, indicating a high level of dissipation of the absorbed light energy into heat, which is consistent with the low effective quantum yield of PSII (Table 2). Considering that some PSII proteins linked to NPQ and Y(NPQ) are involved in the regulation of light absorption by dissipating excess energy as heat through the main and fastest component, qE (energy-dependent component), to avoid photodamage [41], it can be assumed that the phytochrome system and *cry1* are involved in the implementation of this protective mechanism when plants are exposed to UV-B. It can also be assumed that the deficiency of PHYB2 and CRY1 is most critical in the development of oxidative stress induced by UV-B.

One of the main targets of UV-B is the PSII donor oxygen-evolving side [42]; a sufficiently high dose of UV-B results in a reduction in oxygen evolution with the PSII oxygen-evolving complex [40]. Damage to the donor side may be due to the inactivation of D1 and D2 proteins and due to the formation of ROS induced by UV radiation [5,43]. Our data demonstrate a decrease in the expression of the gene encoding the D1 protein in all compared mutants (Figure 1). Apparently, the UV-B-induced decrease in the content of *psbA* transcripts leads to a decrease in the content of the D1 protein and disruption of the PSII donor side. This disorder is especially pronounced in the *phyab2* mutant. This statement is further supported by data on photosynthesis rates, which also decreased in all mutants, but the reduction was the most noticeable in the double and *cry1* mutants (Table 2). However, other components of the electron transport chain are also sensitive to UV-B. For example, UV-B-mediated disruption of PSII pigment-protein intrinsic antenna complexes has been demonstrated [44]. The acceptor side of PSII can also be affected by UV radiation through direct damage to plastoquinone pool molecules [45].

The PSII antenna system (LCH) is composed of LHC family proteins and pigments. Reducing the size of the antenna may serve as a mechanism that protects the photosystem from photoinhibition in strong light [46]. UV-B stress can damage the PSII light-harvesting antenna complex, causing changes in the composition of pigment-binding proteins. This effect can be explained by a decrease in the level of transcription of the *CAB* genes encoding these proteins [26,47]. Indeed, the number of *CAB1* transcripts was markedly reduced in *phyab2* and *phyb1* mutants, which appears to be a protective function to reduce absorbed light under stress conditions, and thus reduce PSII activity and the net photosynthetic rate in these mutants (Figure 1).

Damage to the PSII light harvesting complex may be caused by a decrease in the chlorophyll a/b ratio. Whereas chlorophyll a is found in the core complex of both photosystems (PSI and PSII), chlorophyll b is found in the PSII antenna system. Thus, an increase in the ratio of chlorophyll a/b shows a higher susceptibility of the complex to UV-B compared to the peripheral antenna complex [48]. Before UV-B irradiation, the chlorophyll ratio was slightly higher in the *phyab2* and *cry1* mutants (2.2–2.3) than in the WT (2.1).

Another important protection mechanism against UV-B is the reduction in transpiration [49]. As seen from our data, *phyab2* and *cry1* mutants do not exhibit decreased transpiration, except for *phyb1* (17-fold reduction). The role of CRY1 in the regulation of stomatal aperture is known [50], apparently explaining the lack of influence of UV-B on transpiration rates in the *cry1* mutant (Table 2). Apparently, the same trend could be inherent in PHYB2.

UV-B induces both damage and protection of the PA, stimulating the development of PA-specific defense mechanisms that function through the cooperation of UVR8 with photoreceptors such as cryptochromes and phytochromes [51]. According to our data, these photoreceptors are PHYB2 and CRY1.

However, note that there are some limitations to our observations since we used a model for our study. It is known that the daily dose during the growing period of various crops is small and usually amounts to 2–12 kJ/m^2^ [52]. In our case, the level of UV-B was higher than this dose and the time of action of UV-B was limited, constituting 3 days. Therefore, the adaptation included relatively short-term acting mechanisms. Under natural conditions, even when the proportion of UV-B is significant, only a minor deficit of a photoreceptor might occur. However, to understand the mechanisms of interaction between photoreceptors and UV-B, we used a system that included mutants with a photoreceptor deficit and a high level of UV-B to reliably demonstrate its effects on physiological parameters.

### 2.5. The Role of Light Signaling in UV-B Protection

The effect of moderate UV-B radiation on plant light-signaling genes is mediated through the specific photoreceptor UVR8. When UV-B radiation is absorbed by UVR8, it induces a conformational change that changes the receptor from a dimeric form to a monomeric form. The monomer then interacts with COP1, a key regulator of light signaling. Upon exposure to UV-B, the UVR8 photoreceptor is activated and interacts with the COP1-SUPPRESSOR OF PHYA (SPA) complex, leading to its breakdown [53]. This interaction prevents the interaction of this complex with the transcription factor HY5 and its degradation. This allows HY5 to accumulate and promote the expression of photosensitive genes. An important role in the functioning of UVR8 is played by the proteins RUP1 and RUP2 (repressor of UV-B-induced photomorphogenesis), which in turn contribute to the dimerization of the UVR8 receptor, thereby converting it into an inactive form that absorbs UV-B. Initially, the *phyab2* mutant had increased expression of the *UVR8* and *RUP2* genes; however, after UV-B irradiation, the expression of these genes was markedly reduced, which probably makes this mutant more susceptible to UV-B (Figure 1). After irradiation, the levels of *RUP2, DET1*, and *COP1* transcripts in the *phyab2* mutant were lower than those in the others, which, first, indicates a low ability to convert UVR8 receptors into their original inactive form to reduce cell damage and, second, directly or indirectly involves phytochromes in this mechanism. The expression levels of the *HY5* and *UVR8* genes were also reduced in the *phyab2* mutant upon UV-B irradiation, which is consistent with the low resistance of its PA (Table 2; Figure 1).

DET1 (de-etiolated1) is a key protein involved in the regulation of plant development in response to light. DET1, acting as a negative regulator of photomorphogenesis, helps the COP1/SPA complex maintain low levels of HY5 under these conditions, promoting scotomorphogenesis [54]. DET1 interacts with the COP1/SPA complex, which governs the reaction of UV-B through interactions with UVR8 [55]. When UV-B radiation is low or absent, the COP1/SPA complex targets the degradation of various proteins, including the HY5 transcription factor, which is important for the expression of light-regulated genes. The COP1/SPA complex activates plant protection against UV-B [56], suppressing photomorphogenesis; therefore, the expression of these genes, in general, increased in all plants except for the *phyab2* mutant (Figure 1). In *phyab2* and *cry1* mutants, UV-B caused a two-fold decrease in the expression of *COP1* and *DET1* relative to WT (Figure 1). At the same time, *SPA* transcription was strongly reduced from baseline in all mutants except for *cry1* (Figure 1). The main decrease in *HY5* transcription was observed in the *cry1* mutant. This points to different processes underlying the UV-B sensitivity of these mutants. The *phyab2* mutant showed an initially greater ability to perceive UV-B due to the high transcription of the receptor gene, but under UV-B, the activity was reduced, and the expression of *RUP2* was also decreased, which suggests a reduction mechanism of re-dimerization of UVR8.

We observed a decrease in *HY5* expression in the *cry1* mutant, which suggests the possible involvement of the CRY1 receptor in the COP1/SPA/DET1/HY5 regulatory node (Figure 1). In the work of Ponnu et al. 2019, CRY1 was shown to promote UV-B resistance, enhancing redimerization of the UVR8 receptor [57]. In addition, stabilization of HY5 by CRY by downregulating the COP1/SPA complex results in the expression of genes conferring acclimatization and tolerance to UV-B. Indeed, both CRY and UVR8 are required for plant survival under UV-B conditions, since a triple mutant lacking all *CRY* and *UVR8* genes cannot survive UV-B irradiation [58]. This statement is also true for the tomato plants used in our experiments, since the *cry1* mutant showed reduced resistance to UV-B against the background of a decrease in most of the studied parameters (Table 1 and Table 2). Additionally, it should be noted that repressors of UV-B photomorphogenesis RUP1 and RUP2 interact directly with UVR8 and are also negative regulators of UV-B signaling, inducing the transformation of active UVR8 monomers into inactive dimers [59]. Their expression is decreased in *cry1* and *phyab1* mutants. This is a protective mechanism that allows the mutants to partly improve the negative action of UV-B on PA (Figure 1).

Together with HY5, another positive photomorphogenesis factor, the transcription factor BBX21, is a member of the B-box (BBX) zinc finger protein family and is involved in light signaling in plants [60]. BBX21 has been found to interact directly with HY5 and enhance its activity. Thus, BBX21 could potentially enhance the response to UV-B by stimulating HY5 activity, thereby enhancing the expression of genes for the biosynthesis of secondary metabolites. In our work, we observed an increase in the expression of *BBX21*, especially compared to baseline, and the largest increase was in double and *cry1* mutants (Figure 1). We suggest that, along with a decrease in the expression of *RUP2*-type genes, this may be a compensation mechanism in response to a decrease in the content of *HY5* transcripts, as observed in the *cry1* mutant (Figure 1).

CUL1 and DET1 are parts of the CUL4-DDB1-DET1 E3 ligase complex that targets substrates for proteasomal degradation and suppression of photomorphogenesis [61]. DET1 and CUL1 are negative regulators of light-mediated development and gene expression of positive factors of photomorphogenesis in *Arabidopsis thaliana* [62]. The expression of these genes decreased under UV-B in the *phyab2* mutant (Figure 1). Thus, this protection mechanism from UV-B worked poorly in the *phyab2* mutant (Table 2).

Phytochromomobilin synthase is an enzyme that plays a critical role in the biosynthesis of phytochromomobilin, a chromophore covalently linked to phytochrome apoproteins [63]. This attachment allows phytochromes to absorb and respond to red and far-red light, regulating various aspects of plant development, including seed germination, stem elongation, leaf expansion, and flowering time. Although this enzyme is not directly involved in the perception or signaling of UV-B light, it is an integral part of the function of PHYs that can interact with UVR8 and other components of light-signaling pathways. PHYs and UVR8 interact with the COP1 protein, a key regulator of light signaling. When the plant is exposed to UV-B light, UVR8 is activated and interacts with COP1, disrupting its ability to degrade HY5 and leading to increased expression of light-sensitive genes. Meanwhile, in the presence of red or far-red light, the active form of phytochromes can also interact with COP1, leading to the stabilization of HY5 [64]. Therefore, through the action of phytochromomobilin synthase (*HY2* gene), which functions as a phytochrome, this enzyme indirectly affects the overall response of the plant to light, including the response to UV-B light. In our studies, *HY2* expression was high at the initial point of the experiment in all mutants but decreased after UV-B irradiation (Figure 1). The same can be said for the *PHYB1* and *CRY1* and *CRY2* receptor genes. This further suggests that the plant response to UV-B light is not isolated and may depend on the perception of other wavelengths in the red, far-red, and blue regions. Thus, while phytochromomobilin synthase, PHY, and CRY are not directly involved in UV-B light signaling, they contribute to a complex network of light perception and plant response.

### 2.6. Effect of UV-B and Photoreceptor Deficiency on Pigment Accumulation

In response to certain wavelengths of light, transcription factors provide differential regulation of gene expression for the biosynthesis of various UAPs, primarily flavonoids [65], which can be non-enzymatic antioxidants and protect plants from many stress factors [63]. The main flavonoid compounds present in flowers and fruits are flavonols, anthocyanins, and proanthocyanidins [66]. In our experiments, we observed an increased anthocyanin content compared to the initial level in all mutants, as well as that of UAPs in the WT and *phya* mutant, and a decrease in the content of photosynthetic pigments, the largest of which was in the *cry1* mutant. Moreover, the *phyab2* mutant had the highest contents of anthocyanins and UAPs (Table 1), while the photosynthetic rate and PSII activity were among the lowest (Table 2). On the other hand, *phya* and *phyb1* showed the highest TEAC activity and maintained high effective quantum yields (Table 1 and Table 2). This indicates that *phya* and *phyb1* are more likely to be involved in the biosynthesis of low-molecular weight antioxidants, which act as antioxidants protecting PA from UV-induced oxidative stress rather than shielding PA from UV radiation. This is further supported by the higher expression of *CHS, ANT*, and *PAL* observed in the *phyb1* and *phya* mutants than in the *phyab2* and *cry1* mutants (Figure 1).

Additionally, our results regarding the higher sensitivity of PA to UV radiation in CRY1 and PHYB deficiency agree with the conclusions of a number of works conducted with the use of Arabidopsis mutants, such as *hy4* and *hy3*, with deficits in CRY1 and PHYB, respectively [67,68,69]. Thus, the results obtained in our previous study [67] suggest an important role of phytochrome B in the resistance of Arabidopsis PSII to UV-A radiation and it is suggested that reduced resistance of PSII in the *hy3* mutant with a deficit of phyB is linked to changes in contents of either carotenoids or other UV-absorbing pigments. Additionally, the authors suggested that phytochrome B and other phytochromes can affect PSII stress resistance by the fast regulation of the expression of genes encoding antioxidant enzymes and transcription factors at the step of gene transcription. The Arabidopsis mutant *hy4* with a cry1 deficit showed higher sensitivity to UV-B compared with WT [28]. The authors supposed that reduced resistance of PSII in *hy4* can be associated with low UAP content, as well as lowered POD and CAT enzyme activities. In addition, it was suggested that the lowered expression of *UVR8* and *COP1* genes caused by CRY1 deficiency led to a shift in the balance of oxidants and antioxidants towards oxidants. The same tendency was shown in our data. For example, we indicated lowered levels of the *UVR8* gene in the *phyab2* mutant and the *COP1* gene in the *cry1* and *phyab2* mutants.

The novelty of our work includes some conclusions, especially regarding the key role of PHYB2 and the detailed research into light-inducible genes that are involved in photoreceptor signaling and also important for PA protection from UV-B radiation. Next genes can be attributed to genes such as *UVR8, RUP2, CUL4, ANT, PAL*, and *PHYB2* genes in *phyab2*, and *RUP2, CUL4, ANT, PAL*, and *HY5* genes in the *cry1* mutant.

## 3. Materials and Methods

### 3.1. Plant Materials and Experimental Design

Seed mutants *phya* LA4356, *phyb1* LA4357, *phyab2* LA4362, and *cry1* LA4359, and wild-type (WT) LA2706 tomatoes were obtained from the Tomato Genetics Resource Center, USA, California were chemicaly stratificated. The resulting seedlings were grown under the conditions of the phytotron climatic chamber, 250 µmol (photons) m^−2^s^−1^ (Philips fluorescent lamps TD—L 58 W/33-640 (Pila, Poland) with intensity in the UV-B region 2.3 μmol (photons) m^−2^s^−1^ UV-B (Figure 2), in perlite culture, using Hoagland’s nutrient solution, at a 16 h photoperiod, air temperature of 23/18 °C (day/night), and a humidity of 75%. Plants were grown for 3 weeks and then subjected to vegetative propagation by cuttings. The resulting clones, of 5–7 cm high, were rooted in perlite culture for 1 week and used in the main experiments.

Then, the plants in the climatic chamber were divided into two groups: one group was given an additional 10 µmol (photons) m^−2^s^−1^ UV-B during daylight hours, and the other group remained the control (with 2.7 µmol (photons) m^−2^s^−1^). During the experiment, the plants were additionally irradiated with UV-B 311 nm Phillips (Pila, Poland) ultraviolet lamps (Figure 2) inside the phytotron chamber with a constant air provided outflow by an axial fan (120 × 120 × 25 mm, 12 V) to remove the formed ozone. UV lamps were used as an additional light source, and irradiation was carried out for 16 h with a dark period of 8 h. The duration of the experiment was 72 h, while the total exposure to UV-B during this time was (16 h + 16 h + 16 h) = 48 h.

### 3.2. Pigment Content and Low Molecular Weight Antioxidant Capacity (TEAC)

The contents of chlorophyll *a* (Chl *a*) and *b* (Chl *b*) and the total amount of carotenoids (Car) in the pigment extracts of all studied leaves were determined spectrophotometrically in 80% acetone [70].

The anthocyanin content in all the studied leaves was determined spectrophotometrically in 1% HCl-methanol solution according to Shin et al. 2007.

Low molecular weight antioxidants were extracted with 80% methanol from leaves ground in liquid nitrogen. Low molecular weight antioxidant capacity (Trolox equivalent antioxidant capacity (TEAC)) was determined spectrophotometrically according to the method described by Re et al. 1999 [71].

### 3.3. Photochemical Activity, Transpiration and Net Photosynthetic Rate

Fluorescence induction curves were measured using a mini-PAM fluorometer II (Walz, Effeltrich, Germany) on dark-adapted plants (30 min), as previously described (Klughammer and Schreiber, 2008) [72]. After a pulse of saturating light, plant leaves adapted to 30 min darkness were kept in the dark for one minute and then exposed to actinic light for 5 min followed by pulses of saturating light during which parameters were measured. Blue LEDs (450 nm) were used to produce the measuring light (0.5 μmol photons m^−2^ s^−1^), actinic light (250 μmol (photons) m^−2^ s^−1^, duration 10 min), and saturating pulses (450 nm, μmol photons m^−2^ s^−1^ and duration 800 ms). Parameters based on fluorescence data were determined using Imaging Win v.2.41a software (Walz, Germany). Values for F_0_, F_v_, F_m_, F_m_’, and F_0_’ were determined. F_m_ and F_m_’ are the maximum levels of chlorophyll fluorescence under dark- and light-adapted conditions, respectively. F_v_ is the photoinduced change in fluorescence, and F_t_ is the level of fluorescence before the saturation pulse was applied. F_0_ is the initial level of chlorophyll fluorescence. Based on these results, the maximum (F_v_/F_m_) and effective Y(II) (F_m_’ − F_t_)/F_m_’ PS II photochemical photon yields and non-photochemical quenching (NPQ) (F_m_/F_m_’ − 1) were determined. We also determined the values of Y(NO) and Y(NPQ) quantum yields of non-regulated and regulated non-photochemical energy dissipation in PSII, respectively.

The photosynthetic and transpiration rates were determined using a portable infrared gas analyzer (CIRAS-2 PP systems, Haverhill Road, MA USA), which was connected to a 2.5 cm^2^ chamber. The intensity of the light flux corresponded to that used in the climatic chamber and amounted to 250 μmol (photons) m^−2^s^−1^, and the CO_2_ content in the measurement chamber was 400 ppm.

### 3.4. RNA Extraction and RT-PCR

RNA isolation from tomato leaves was performed using TRIzol (Sigma, Burlington, MA, USA). The quantity and quality of total RNA were determined using a NanoDrop 2000 spectrophotometer (Thermo Fisher Scientific, Waltham, MA, USA). cDNA synthesis was performed using the M-MLV reverse transcriptase kit (Fermentas, Waltham, MA, USA), primer oligo (dT) 21 for nuclear genes, and universal primer Random 6 for chloroplast genes. Gene expression patterns were assessed using the CFX96 Touch™ real-time PCR detection system (Bio-Rad, Hercules, CA, USA). Gene-specific primers (Appendix A) chalconesynthase (*CHS*, NM_001247104.2), elongated hypocotyl 5 (*HY5*, NM_001247891.2), phytochrome interacting factor 4 (*PIF4*, NM_001308008.1), phenylalanine ammonia-lease (*PAL1*, XM_004249510.4), cullin 4 (*CUL4*, EU218537.1), suppressor of PHYA protein (*SPA*, NM_001320396.1), B-box protein 21 (*BBX21*, XM_004238269.4), phytochromobilin synthase (*HY2*, A0A3Q7E952), cryptochrome 1 (*CRY1*, Solyc04g074180) cryptochrome 2 (*CRY2*, Solyc09g090100), anthocyanidin synthase (*ANT*, NM_001374394.1), S subunit PSII (*PSBS*, Solyc06g060340.3.1), chlorophyll a/b binding protein 1 (*CAB1*, AH001371.2), photosystem II protein D1 (*psbA*, YP_008563068.1), photosystem II protein D2 (*psbD*, YP_008563083.1), de-etiolated 1 (*DET1*, AJ224356.1), E3 ubiquitin-protein ligase (*COP1*, AY842290.1), phytochrome B1 (*PHYB1*, NM_001306202.1), ultraviolet—B receptor (*UVR8*, XM_019214123.2), and repressor of UV-B photomorphogenesis 2 (*RUP2*, XM_015201575.2) (S.1) were selected using nucleotide sequences from the National Center for Biotechnology Information (NCBI) database (www.ncbi.nlm.nih.gov, USA accessed on 1 February 2023), https://www.uniprot.org/ accessed on 1 February 2023, and https://phytozome-next.jgi.doe.gov/ accessed on 1 February 2023, with Vector NTI Suite 9 software (Invitrogen, Carlsbad, CA, USA). Transcript levels were normalized according to *Tubulin 1* gene expression. Gene expression in WT was assigned a value of 1.

### 3.5. Statistics

Fluorescence measurements were taken, and photosynthetic and transpiration rates determined, in four to six biological replicates on the developed leaves of the middle tier. Each plant fixed in liquid nitrogen was treated as a biological replicate; thus, three biological replicates were performed to determine the pigments, TEAC, anthocyanins, and UAPs, and to analyze the gene expression. For each of these experiments, at least three parallel independent measurements were taken. The significance of differences between groups was calculated using one-way analysis of variance (ANOVA) followed by Duncan’s method using SigmaPlot 12.3 (Systat Software Inc., Inc., San Jose, CA, USA). Letters indicate significant differences between mutants (*p* < 0.05) unless otherwise specified. Data are given as arithmetic means ± standard errors.

## 4. Conclusions

We have shown that the PA of *cry1* and *phyab2* mutants was the most susceptible to UV-B exposure, but this phenomenon is based on different mechanisms. The low resistance of the *phyab2* mutant is probably associated not only with the direct or indirect involvement of PHYB2 in light signaling under UV-B conditions, but also with lower levels of expression of the *UVR8* and *RUP2* genes; the latter activates the formation of the UVR8 dimer absorbing UV-B. A decrease in the content of the dimers does not allow sufficient activation of the defense systems associated with the UVR8 photoreceptor, which ultimately leads to a decrease in PA resistance. The *cry1* mutant is also less resistant, but in this case, UV-B irradiation significantly affects the expression of light-sensitive genes necessary for light signaling and photomorphogenesis, such as *HY5* and *RUP2*, which ultimately leads to a decrease in the value of TEAC and, hence, PA resistance.

## Figures and Tables

**Figure 1 ijms-24-13142-f001:**
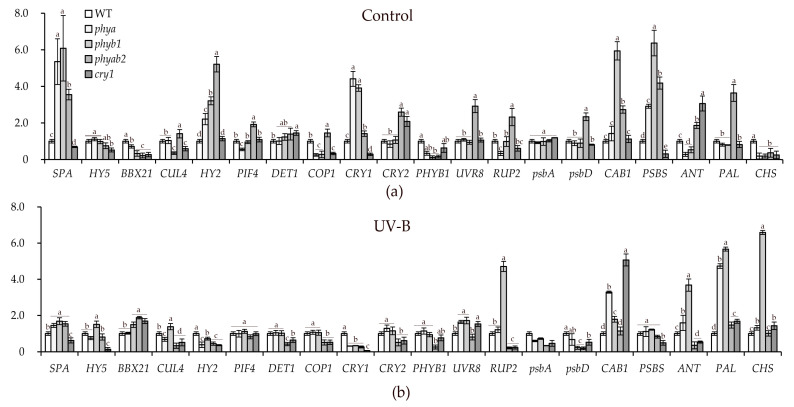
Transcript levels of phenylalanine ammonia-lyase (*PAL)*, chalcone synthase (*CHS)*, elongated hypocotyl 5 (*HY5)*, phytochrome interacting factor 4 (*PIF4*), cullin 4 (*CUL4*), suppressor of PHYA protein (*SPA*), b-box transcriptional facrot 21 (*BBX21)*, phytochromobilin synthase (*HY2)*, cryptochrome 1 (*CRY1)*, cryptochrome 2 (*CRY2)*, anthocyanin synthase (*ANT)*, S subunit of PSII (*PSBS)*, chlorophyll a/b binding protein 1 (*CAB1)*, photosystem II protein D1 *(psbA)*, photosystem II protein D2 (*psbD)*, de-etiolated1 (*DET1*), E3 ubiquitin-ligase (*COP1*), phytochrome B1 (*PHYB1*), ultraviolet-B receptor (*UVR8*), and repressor of UV-B photomorphogenesis 2 (*RUP2*) under white fluorescent lamps (**a**) and under UV-B (311 nm) after 48 h of treatment (**b**). The transcript levels were normalized to the expression of the *Tubulin1* gene. The gene expression in WT was used as one unit. Different letters indicate significant differences (*p* ≤ 0.05) between the experimental treatments for each gene, *n* = 3.

**Figure 2 ijms-24-13142-f002:**
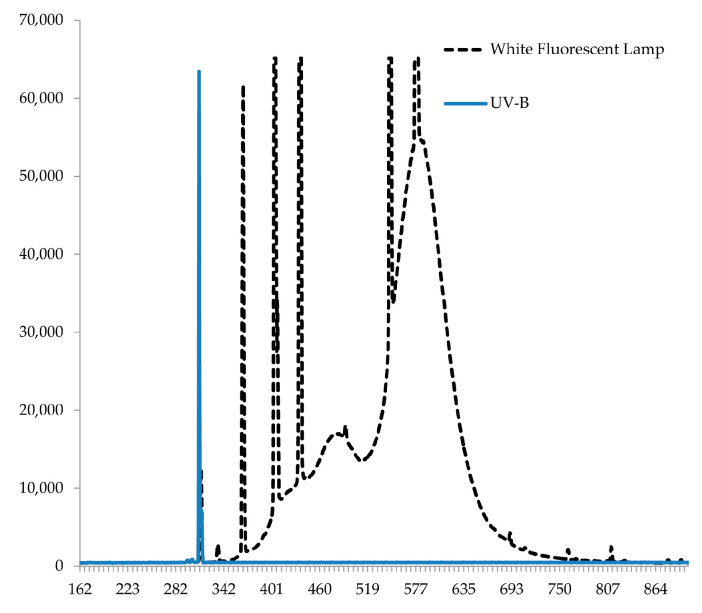
Spectrogram of light sources used in experiments.

**Table 1 ijms-24-13142-t001:** The content of the main photosynthetic pigments (chl *a*, chl *b*, car mg g^−1^, DW), anthocyanins (µg g^−1^, FW), and TEAC activity (Trolox equivalent antioxidant capacity, µmol^−1^ Trolox, FW) in the leaves of tomato plant mutants for the main photoreceptors within 48 h.

	Control				
	WT	*phya*	*phyb1*	*phyab2*	*cry1*
Chl *a*	8.3 ± 1.3 ab	8.6 ± 1.6 a	8.3 ± 1.6 a	7.2 ± 0.9 b	9.6 ± 1.6 a
Chl *b*	4.0 ± 0.6 a	4.1 ± 1.2 a	4.0± 0.6 a	3.2 ± 0.4 b	4.3 ± 0.6 a
Car	1.55 ± 0.16 a	1.11 ± 0.35 ab	1.11 ± 0.21 ab	1.10 ± 0.13 b	1.30 ± 0.26 ab
Chl *(a* + *b)*	12.3 ± 1.9 a	12.7 ± 2.8 a	12.3 ± 2.2 a	10.4 ± 1.3 b	13.9 ± 2.2 a
Anthocyanins	1.09 ± 0.04 a	0.55 ± 0.02 c	0.36 ± 0.04 d	0.81 ± 0.02 b	0.54 ± 0.13 c
UAPs	6.6 ± 0.2 c	6.6 ± 0.2 c	16.6 ± 0.5 a	10.3 ± 0.2 b	7.8 ± 2.4 c
TEAC	17.91 ± 0.20 c	19.73 ± 0.29 a	15.03 ± 0.15 e	18.73 ± 0.12 b	16.27 ± 0.21 d
	UV-B				
	WT	*phya*	*phyb1*	*phyab2*	*cry1*
Chl *a*	4.9 ± 1.2 b	5.4 ± 0.7 b	5.2 ± 0.7 b	6.6 ± 0.8 a	4.3 ± 0.6 b
Chl *b*	2.3 ± 0.6 b	2.5 ± 0.3 b	2.5 ± 0.3 b	3.2 ± 0.4 a	2.1 ± 0.3 b
Car	0.62 ± 0.24 b	0.65 ± 0.10 b	0.66 ± 0.08 b	0.86 ± 0.12 a	0.65 ± 0.09 b
Chl *(a* + *b)*	7.2 ± 1.8 b	7.9 ± 1.0 b	7.7 ± 1.0 b	9.8 ± 1.2 a	6.4 ± 0.9 c
Anthocyanins	2.69 ± 0.08 a	1.55 ± 0.19 bc	0.42 ± 0.07 e	2.02 ± 0.26 b	0.93 ± 0.07 c
UAPs	10.1 ± 1.0 ab	9.0 ± 0.7 b	9.9 ± 0.9 ab	11.2 ± 0.9 a	8.1 ± 0.8 b
TEAC	19.37 ± 0.12 c	24.23 ± 0.06 a	22.01 ± 0.20 b	17.73 ± 0.12 d	16.21 ± 0.55 e

Different letters indicate significant differences (*p* ≤ 0.05) according to ANOVA on ranks followed by Duncan’s method. WT = WT; Chl = chlorophyll; Car = carotenoids; UAPs = ultraviolet adsorption pigments.

**Table 2 ijms-24-13142-t002:** Effect of UV-B on transpiration (Tr, mmol H_2_O m^−2^s^−1^) and net photosynthetic rates (Pn, μmol m^−2^s^−1^) and the main parameters of chlorophyll *a* fluorescence: Y(II) (PSII effective quantum yield), NPQ (non-photochemical fluorescence quenching), Y(NO) (quantum yield of non-regulated non-photochemical energy dissipation in PSII), Y(NPQ) (quantum yield of regulated non-photochemical energy dissipation in PSII) in the leaves of tomato photoreceptor mutant plants after 48 h of treatment.

	Control				
	WT	*phya*	*phyb1*	*phyab2*	*cry1*
Fv/Fm	0.841 ± 0.004 a	0.852 ± 0.003 a	0.830 ± 0.009 a	0.843 ± 0.001 a	0.850 ± 0.005 a
Y(II)	0.33 ± 0.01 a	0.38 ± 0.04 a	0.35 ± 0.04 a	0.33 ± 0.03 a	0.35 ± 0.01 a
NPQ	1.45 ± 0.03 a	0.99 ± 0.10 c	1.28 ± 0.06 b	1.28 ± 0.04 b	1.26 ± 0.03 b
Y(NO)	0.27 ± 0.01 a	0.31 ± 0.01 a	0.29 ± 0.01 a	0.29 ± 0.01 a	0.29 ± 0.01 a
Y(NPQ)	0.40 ± 0.01 a	0.33 ± 0.04 a	0.37 ± 0.02 a	0.36 ± 0.04 a	0.36 ± 0.01 a
Tr	0.60 ± 0.20 c	1.30 ± 0.10 b	1.70 ± 0.40 a	0.30 ± 0.10 d	0.30 ± 0.10 d
Pn	6.1 ± 0.5 a	4.5 ± 0.3 b	6.4 ± 0.9 a	4.8 ± 0.3 b	3.9 ± 0.3 c
	UV-B				
	WT	*phya*	*phyb1*	*phyab2*	*cry1*
Fv/Fm	0.832 ± 0.009 b	0.803 ± 0.032 c	0.842 ± 0.002 a	0.833 ± 0.015 b	0.834 ± 0.013 b
Y(II)	0.24 ± 0.01 b	0.31 ± 0.01 a	0.25 ± 0.02 b	0.15 ± 0.03 c	0.15 ± 0.02 c
NPQ	0.24 ± 0.02 b	0.31 ± 0.01 a	0.25 ± 0.03 b	0.26 ± 0.02 b	0.26 ± 0.04 ab
Y(NO)	0.29 ± 0.01 b	0.27 ± 0.03 b	0.32 ± 0.03 b	0.31 ± 0.02 b	0.53 ± 0.03 a
Y(NPQ)	0.48 ± 0.03 ab	0.41 ± 0.03 b	0.39 ± 0.03 b	0.53 ± 0.04 a	0.36 ± 0.03 b
Tr	0.11 ± 0.04 c	0.50 ± 0.11 a	0.10 ± 0.09 c	0.50 ± 0.06 a	0.31 ± 0.09 b
Pn	3.7 ± 0.3 a	3.4 ± 0.4 a	2.8 ± 0.6 ab	1.4 ± 0.2 b	0.9 ± 0.3 c

Different letters indicate significant differences (*p* < 0.05) between the experimental treatments.

## Data Availability

The datasets generated and/or analyzed during the current study are available from the corresponding author upon reasonable request.

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
