# Peer review of "Potential Role of Phytochromes A and B and Cryptochrome 1 in the Adaptation of Solanum lycopersicum to UV-B Radiation"

_ijms, 2023, doi:10.3390/ijms241713142_

Round 1

Reviewer 1 Report

The paper „Potential Role of Phytochromes A and B and Cryptochrome 1 in 2

the Adaptation of Solanum lycopersicum to UV-B Radiation” describes the impact of UVB irradiation on photosynthetic parameters, antioxidant capacity, levels of photosythetic pigments  and transcripts of photosynthesis-related and light-signal related genes in a set of tomato cryptochrome 1 and phytochrome mutants. The Authors found that tomato response to UVB had depended on the action of PHYB2 and CRY1 photoreceptors.

The presented results are interesting, however the paper needs some improvements:

- all the results should be presented as graphs, not only tables. It would be more reader friendly.

- the standard deviation of the mean level of the gene expressions in WT plants should be added (even if „gene expression in WT was assigned a value of 1”).

- which gene was used as a reference in the real-time PCR measurements

- what does „nuclear gene of Suppressor of PHYA protein for chloroplast product” mean? Should it be just ”suppressor of PHYA protein”?

- the numer of biological, independent replicates used for each experiments should be added. How many plants were tested in each replicate?

- which part of the plant was collected for RNA isolation?. How many plants were used for the experiments? What does n=3 mean?

- Was the UVB irradiation uniform? Did each plant genotype obtain the same UVB dose?

Author Response

  1. All the results should be presented as graphs, not only tables. It would be more reader friendly.

Answer:

We are grateful to the reviewer for suggestion. We agree that data is easier to read in graph format. We have converted table whith genes expression into graphs.

  1. The standard deviation of the mean level of the gene expressions in WT plants should be added (even if „gene expression in WT was assigned a value of 1”).

Answer:

We are grateful to the reviewers for their suggestion it is done.

  1. Which gene was used as a reference in the real-time PCR measurements

Answer: We are grateful to the reviewers for the discovered inaccuracy. Tubulin 1 was used as a reference gene, this phrase was added to Table 3 captions and materials and methods section.

  1. What does „nuclear gene of Suppressor of PHYA protein for chloroplast product” mean? Should it be just ”suppressor of PHYA protein”?

Answer: We are grateful to the reviewers for the discovered inaccuracy. It is corrected.

  1. The numer of biological, independent replicates used for each experiments should be added. How many plants were tested in each replicate?

Answer:

We are grateful to the reviewers for the discovered inaccuracy.

This frase was added to 4.5. Statistics section «Fluorescence measurements, photosynthetic and transpiration rates were carried out in four to six biological replicates on developed leaves of the middle tier. Each plant fixed in liquid nitrogen was treated as a biological replicate; thus, three biological replicates were performed to determine the pigments, TEAC, anthocyanin, UAPs and to analyse gene expression. For each of these experiments, at least three parallel independent measurements were made.»

  1. Which part of the plant was collected for RNA isolation? How many plants were used for the experiments? What does n=3 mean?

Answer:

Only developed leaves of the middle tier were used in the analysis. At least three plants were used at each analysis point. n - is the number of independent biological replicates.

  1. Was the UVB irradiation uniform? Did each plant genotype obtain the same UVB dose?

Answer:

Thank you for the question. We attempted to ensure an even UVB dose for each plant as much as possible. To achieve this, we used a setup with six lamps. Additionally, since the tomato is a longline plant, the topmost leaves might incur more damage, while the middle leaves might receive a lesser dose. We sought to minimize this effect by conducting multiple repetitions of our experiments, including pilot tests.

Reviewer 2 Report

The research article investigates the intricate interactions between photoreceptors and the protective mechanisms activated in tomato plants under UV-B irradiation. While the effects of UV-B on photosynthetic apparatus damage and regulatory responses are well-known, the specific components of these interactions remain unclear. This study employs tomato photoreceptor mutants (phya, phyb1, phyab2, cry1) to delve into these interactions. This work has the potential to influence future research in plant photobiology and UV-B stress responses. In conclusion, the paper would be sufficient to merit publication in IJMS though a major revision is recommended which needs to include the following points.

(1) The presentation of results is not good. The results should be presented in an easy-to-understand manner, using graphs, etc. Also, the explanation of the results is insufficient. Providing detailed explanations of the observed changes in the context of photoreceptor interactions would enhance the reader's understanding.

(2) I don't know what Figure 2 shows. Nor can I determine that it is supported by the results. It is misleading and should be removed.

(3) The article could benefit from a discussion of potential limitations, such as the specific conditions of UV-B exposure and the possible implications of using mutants. Addressing these limitations would strengthen the overall validity of the findings.

(4) Comparing the current findings to existing literature on plant responses to UV-B irradiation would contextualize the significance of the results and highlight novel insights provided by the study.

Author Response

  1. The presentation of results is not good. The results should be presented in an easy-to-understand manner, using graphs, etc. Also, the explanation of the results is insufficient. Providing detailed explanations of the observed changes in the context of photoreceptor interactions would enhance the reader's understanding.

Answer:

We are grateful to the reviewer for suggestion. We agree that data is easier to read in graph format. We have converted table whith genes expression into graphs. Also, we improved the explanation of the results.

  1. I don't know what Figure 2 shows. Nor can I determine that it is supported by the results. It is misleading and should be removed.

Answer:

Thanks for the comment, we crossed out the Figure 2.

  1. The article could benefit from a discussion of potential limitations, such as the specific conditions of UV-B exposure and the possible implications of using mutants. Addressing these limitations would strengthen the overall validity of the findings.

Answer:

It is done.

The text was added to 2.2. Photosynthetic activity, net photosynthetic and transpiration rates Section  «Initially The portion of UV-B available in the spectrum irradiation had no significant effect on the PSII activity and photosynthesis rate of the WT (Table 2), which corresponded to corresponding values indicated earlier in pine seedlings (Pashkovskiy et al. 2021). Such level of UV-B likely affects the photosynthetic parameters not so much. Hence, for demonstration of significant effects of UV-B we needed to use higher doses of UV-B. The application of added 10 µmol (photons) m-2s-1 for 1 and 2 h  did not lead to inhibition of photosynthetic activity but irradiation with UV-B for 16 h led to small but reliable effect on PA activity (data not shown). However, irradiation for 3 days led to significant influence of added UV-B on different activities.»

  1. Comparing the current findings to existing literature on plant responses to UV-B irradiation would contextualize the significance of the results and highlight novel insights provided by the study.

Answer:

It is done.

The text was added to 3.1. Impact of UV-B and photoreceptor deficiency on photosynthetic processes Section:

«However, note that there are some limitations to our observations since we used a model for our study. It is known that the daily dose during the growing period of various crops is small and usually amounts to 2–12 kJ/m2 (Kakani et al. 2003). In our case the level of UV-B is higher than this dose. In addition, the time of action of UV-B was limited and constituted 3 days. Therefore, the adaptation includes relatively short-term acting mechanisms. Also, under real nature conditions when the portion of UV-B is significant only small deficit of a photoreceptor is possible but for the study of comprehencion of mechanisms of interaction between photoreceptors and UV-B we had to use the system including mutants with photoreceptor deficit and high level of UV-B to indicate its reliable effects on physiological parameters»

And to 3.3. Effect of UV-B and photoreceptor deficiency on pigment accumulation SectionSection « Also, our results about higher sensitivity of PA to UV-radiation at cry1 and phyB deficiency are agreed with the conclusions of a number of works conducted with the use of Arabidopsis mutants such as hy4 and hy3 with deficit of cry1 and phyB, respectively (Kreslavski et al. 2916; 2018; Khudyakova et al. 2019). Thus, the results obtained in our previous study (Kreslavski et al. 2016) suggest an important role phytochrome B in the resistance of Arabidopsis PSII to UV-A radiation and it is suggested that reduced resistance of PSII in hy3 mutant with deficit of phyB is linked to changes in contents of either carotenoids or other UV-absorbing pigments. Also, the authors suggested that phytochrome B and other phytochromes can affect the PSII stress resistance by the fast regulation of the expression of genes encoding antioxidant enzymes and transcription factors at the step of gene transcription. Arabidopsis mutant hy4 with cry1 deficit showed higher sensitivity to UV-B compared with WT (Khudyakova et al. 2019). The authors supposed that that reduced resistance of PSII in hy4 can be associated with low UAPs content as well as lowered POD and CAT activities. In addition, it is suggested the lowered expression of UVR8 and COP1 genes caused by Cry1 deficiency leads to a shift of balance of oxidants and antioxidants towards oxidants. The same tendency showed or data. For example, we indicated lowered levels of uvr8 gene in phyab2 mutant and COP1 gene in cry1 and phyab2 mutants.

Novelty of our work is some conclusions, especially on key role of phyB2 and detailed research of light-indicible genes which are involved in photoreceptor signaling and also important for PA protection from UV-B radiation. Next genes can be attributed to such genes: UVR8, RUP2, CUL4, ANT, PAL and PHYB2 genes in phyab2 and RUP2, CUL4, ANT, PAL and HY5 genes in the cry1 mutant.»

Round 2

Reviewer 1 Report

Please, check the English. Be more precise, especially in the Results section. Some details are given below:

The Authors state: „Despite this, the question of what is the role of photoreceptors that are not directly involved in the reception of light of various spectral compositions remains insufficiently studied.”. But it is not the issue of the paper. The plants were cultured under white light! (supplemented with UVB), thus photoreceptors including these activated by blue, red and far red spectra triggered the response. These responses could help plant to cope with UV-induced damage.

Lines 109-110 should be „role in UV-B …” instead of „ role in and UV-B…”

Correct the English. In line 16, what does mean: „pathways of this interaction”. Which interaction? In the previous sentence, “interactions” are in the plural. The study focus on the putative role of photoreceptors in plant responses/sensitivity to UV-B, I cannot find any data on any interaction(s).  

Anthocyanin should be in plural (for example see line: 19).

Line 85: what does “protein genes” mean?

Line 93: should it be: MYBs?

Please, explain the abbreviation on the first use (see for example: ANT)

Line 115 what does : ”double mutant” mean?

Line 119: what does : ”variants” mean?

Please, correct the Results section, for example: “content was maximum”-  it is not in English. Simplify the descriptions and be more precise. For example: “The UAPs content was maximum in the phyb1 mutant, while in phyab2, it was 1.6 times lower but more than in other variants. However, after exposure to UV-B, this difference was not as significant. It was maximal between phyab2 and cry1, whose UAPs content was 1.5 times lower than that of phyab2”. Which difference should be significant? What was maximal?, what does “between phyab2 and cry1” mean?

Line 141, what does “spectrum irradiation” mean? The Authors state: “Initially the portion of UV-B available in the spectrum irradiation had no significant effect on the PSII activity and photosynthesis rate of the WT” – what was the control? That means it has no effect in comparison to which conditions?

Line 149, what does “different activities” mean?

Please check the citations. I am not able to find that „6% of UV-A (315–400 nm) and less than 0.5% of UV-B (280–315 nm)” in the paper of Favory et al.  2009.

All the details concerning the numer of biological replicates, the plant parts used for the experiments and so on (as given in response to the Reviewers) should be added to m&m section.  

Please, check the English. The details are given in the above section.

Author Response

1. Please, check the English. Be more precise, especially in the Results section. Some details are given below:

Answer: We improve the English throughout the text.

2. The Authors state: „Despite this, the question of what is the role of photoreceptors that are not directly involved in the reception of light of various spectral compositions remains insufficiently studied.”. But it is not the issue of the paper. The plants were cultured under white light! (supplemented with UVB), thus photoreceptors including these activated by blue, red and far red spectra triggered the response. These responses could help plant to cope with UV-induced damage.

Answer: Dear Reviewer, thank you for pointing out this discrepancy. We recognize the concerns raised regarding our statement about photoreceptors that are not directly involved in the reception of light of various spectral compositions.

Our intention behind the statement was to emphasize the broader scientific context and acknowledge the gaps in our understanding of photoreceptors outside of their primary spectral targets. While our experiment did use white light supplemented with UVB, and thus activated a range of photoreceptors including those sensitive to blue, red, and far-red light, the core focus of our study was to understand the interplay between UVB and plant physiological responses.

In light of your comment, we can see how our statement might have been misleading or confusing. We will clarify this in the revised manuscript to ensure that the reader is not left with an impression that we are addressing the roles of all photoreceptors in this study. Instead, we will emphasize our study's focus on the specific effects of UVB supplementation and how various photoreceptors, activated by white light, might aid in modulating UV-induced damages.

We improved the last part of Introduction section: «Under natural environmental conditions, sunlight activates all photoreceptors, because it contains an extremely wide spectrum of radiation. At the same time, the spectral composition of light can differ at sunrise and sunset, and intensity of the ultraviolet radiation is related to the intensity of sunlight, and the height above sea level where the plants grow. In this regard, we studied photoreceptor mutants to understand how they are involved in light signaling under conditions of additional long-term UV-B irradiation.»

3. Lines 109-110 should be „role in UV-B …” instead of „ role in and UV-B…”

Answer: It is done

4. Correct the English. In line 16, what does mean: „pathways of this interaction”. Which interaction? In the previous sentence, “interactions” are in the plural.The study focus on the putative role of photoreceptors in plant responses/sensitivity to UV-B, I cannot find any data on any interaction(s).  

Answer: We agree and changed the text : However, a role of the photoreceptors in the responses of the plants to UV-B remains undiscovered. This study explores some of these responses using tomato photoreceptor mutants (phya, phyb1, phyab2, cry1).

5. Anthocyanin should be in plural (for example see line: 19).

Answer: It is done

6. Line 85: what does “protein genes” mean?

Answer: It is done

7. Line 93: should it be: MYBs?

Answer: It is done

8. Please, explain the abbreviation on the first use (see for example: ANT)

Answer: It is done.

9. Line 115 what does : ”double mutant” mean?

Answer: We change it to phyab2.

10. Line 119: what does : ”variants” mean?

Answer: «Variants» mean «mutants», we improve text.

11. Please, correct the Results section, for example: “content was maximum”-  it is not in English. Simplify the descriptions and be more precise. For example: “The UAPs content was maximum in the phyb1 mutant, while in phyab2, it was 1.6 times lower but more than in other variants. However, after exposure to UV-B, this difference was not as significant. It was maximal between phyab2 and cry1, whose UAPs content was 1.5 times lower than that of phyab2”. Which difference should be significant? What was maximal?, what does “between phyab2 and cry1” mean?i

Answer It is done.

We corrected the Results section and wrote:

  1. Initially, the anthocyanins content was highest in the WT and lowest in the cry1 mutant.
  2. The UAPs content was the highest in the phyb1 mutant, while in phyab2 it was 1.6 times lower but higher than in other mutants. However, after exposure to UV-B, this difference among mutants was negligible. Together with that, the UAPs content in cry1 was 1.5 times lower than that in phyab2”.
  3. Initially, TEAC value was the highest in the phya mutant and the lowest in phyb1 (1.3 times lower than in the phya). After UV-B irradiation, the highest TEAC was observed in phya, while the TEAC in phyab2 and cry1 mutants was 1.4-1.5 times lower than in the phya. The phyab2 and phyb1 mutants showed intermediate TEAC values (Table 1).

12. Line 141, what does “spectrum irradiation” mean? The Authors state: “Initially the portion of UV-B available in the spectrum irradiation had no significant effect on the PSII activity and photosynthesis rate of the WT” – what was the control? That means it has no effect in comparison to which conditions?

Answer: 

We corrected and wrote "irradiation" instead of "spectrum irradiation"

What was the control? The control was wild type before and after UV-B irradiation.

That means it has no effect in comparison to which conditions? This means that initially the plants differed little from the initial point of the experiment, and then differences appeared. The phrase is misleading we changed it.

13. Line 149, what does “different activities” mean?

Answer: We wrote: «activity of PSII and net photosynthetic rate»

14. Please check the citations. I am not able to find that „6% of UV-A (315–400 nm) and less than 0.5% of UV-B (280–315 nm)” in the paper of Favory et al.  2009.

Answer: It is done. We changed the reference.

Palma, C.F.F.; Castro-Alves, V.; Morales, L.O.; Rosenqvist, E.; Ottosen, C.-O.; Strid, Å. Spectral Composition of Light Affects Sensitivity to UV-B and Photoinhibition in Cucumber. Front Plant Sci 2021, 11, 610011, doi:10.3389/fpls.2020.610011

15. All the details concerning the numer of biological replicates, the plant parts used for the experiments and so on (as given in response to the Reviewers) should be added to m&m section.

Answer: We improve the information in MM section in subsection Statistics.

4.5. Statistics

Fluorescence measurements, photosynthetic and transpiration rates were carried out in four to six biological replicates on developed leaves of the middle tier. In all ana-lyzes, only leaves of the middle tier were studied. For determine the pigments, TEAC, anthocyanins, UAPs and to analyse gene expression three biological replicates were performed. For each of these experiments, at least three parallel independent measurements were made. The significance of differences between groups was calculated using one-way analysis of variance (ANOVA) followed by Duncan's method using SigmaPlot 12.3 (SystatSoftwareInc., USA). Letters indicate significant differences between variantsmutants (p < 0.05) unless otherwise specified. Data are given as arithmetic means ± standard errors.

Reviewer 2 Report

The authors have properly revised the paper in accordance with my comments.

Author Response

Many thanks to the reviewer for appreciating our work.

Round 3

Reviewer 1 Report

The paper was corrected thoroughly. In my opinion it can be accepted in the present form.